# Association of the KDIGO Risk Classification with the Prevalence of Heart Failure in Patients with Type 2 Diabetes

**DOI:** 10.3390/jcm10204634

**Published:** 2021-10-09

**Authors:** José Antonio Gimeno-Orna, Luis Rodríguez-Padial, Manuel Anguita-Sánchez, Vivencio Barrios, Javier Muñiz, Antonio Pérez

**Affiliations:** 1Endocrinology and Nutrition Service, Hospital Clínico Universitario Lozano Blesa, 50009 Zaragoza, Spain; 2Cardiology Service, Complejo Hospitalario de Toledo, 45004 Toledo, Spain; lrpadial@gmail.com; 3Cardiology Service, Hospital Universitario Reina Sofía, Instituto Maimónides de Investigación Biomédica (IMIBIC), Universidad de Córdoba, 14004 Córdoba, Spain; manuelanguita@secardiologia.es; 4Cardiology Service, Hospital Universitario Ramón y Cajal, 28034 Madrid, Spain; vivenciobarrios@gmail.com; 5Coruña University, Cardiovascular Research Group, Health Sciences Department and Biomedic Research Institute de A Coruña (INIBIC), CIBERCV, 15006 A Coruña, Spain; javmu@odds.es; 6Endocrinology and Nutrition Service and Research Institute (IIB Sant Pau) of the Hospital de la Santa Creu i Sant Pau, Universidad Autónoma de Barcelona, CIBER de Diabetes y Enfermedades Metabólicas (CIBERDEM), 08023 Barcelona, Spain

**Keywords:** type 2 diabetes, cardiovascular disease, heart failure, KDIGO risk category

## Abstract

The objectives of this study were to determine the main characteristics associated with the presence of heart failure (HF) in patients with type 2 diabetes (T2DM), and specifically to assess the association of the risk classification proposed by the Kidney Disease Improving Global Outcomes (KDIGO) guidelines with HF. The DIABET-IC study is a multicentre, observational, prospective and analytical study in T2DM patients recruited in Spanish hospitals. This work, which features a cross-sectional design, has been conducted with the data obtained at the inclusion visit. The main dependent variable analysed was the presence of HF. The predictive variables evaluated were the demography, clinic, laboratory testing (including natriuretic peptides) and echocardiography. Patients were classified according to the number of vascular territories with atherosclerotic involvement and the KDIGO risk category. Multivariate logistic regression models were performed to determine the risk posed by the various baseline variables to present HF at the time of study inclusion. The study included 1517 patients from 58 hospitals, with a mean age of 67.3 (standard deviation (SD): 10) years, out of which 33% were women. The mean DM duration was 14 (SD: 11) years. The prevalence of HF was 37%. In a multivariate analysis, the independent predictors of HF were increased age (odds ratio (OR) per 1 year = 1.02; *p* = 0.006), decreased systolic blood pressure (OR per 1 mmHg = 0.98; *p* < 0.001), decreased haemoglobin (OR per 1 g/dL = 0.86; *p* < 0.001), the presence of obstructive sleep apnoea (OR = 1.61; *p* = 0.006), the absence of hepatic steatosis (OR = 0.59; *p* = 0.016), the severity of atherosclerotic involvement (OR 1 territory = 1.38 and OR > 1 territory = 2.39; *p* = 0.02 and *p* < 0.001 respectively) and the KDIGO risk classification (high-risk OR = 2.46 and very high-risk OR = 3.39; *p* < 0.001 for both). The KDIGO risk classification is useful to screen for the presence of HF in T2DM patients. Therefore, we believe that it is necessary to carry out a systematic screening for HF in the high- and very high-risk KDIGO categories.

## 1. Introduction

It is widely known that patients with diabetes have an increased risk of cardiovascular disease (CVD) [1]. The presence of diabetes mellitus increases the risk of heart failure (HF) by more than 50%. At present, HF is the second most frequent manifestation of CVD in patients with type 2 diabetes mellitus (T2DM) [2]. In a large population-based study, the most important risk factors identified for HF hospitalization in T2DM patients were atrial fibrillation, excess body weight and the loss of renal function [3]. In fact, despite having HbA1c levels, blood pressure, LDL-cholesterol and urinary albumin excretion within the recommended targets, the increased risk given by T2DM is not completely neutralised.

Pathophysiologically, the increased risk of HF in T2DM patients can be attributed to the presence of ischemic heart disease (myocardial infarction being a main cause of HF with reduced ejection fraction), high blood pressure and the presence of a specific diabetic cardiomyopathy [4]. Insulin resistance, hyperglycaemia, myocardial lipotoxicity with mitochondrial dysfunction and oxidative stress, and the inflammatory response lead to myocardial fibrosis and diastolic dysfunction [5]. Myocardial microvascular dysfunction is considered to play a key role in the development of HF with preserved ejection fraction [6].

The development of diabetic chronic kidney disease (CKD) is an additional factor that increases the likelihood of HF, mainly because it facilitates the development of atherosclerotic lesions, fibrosis, left ventricular hypertrophy and valvular calcifications [7]. The 2012 Guidelines of the Kidney Disease Improving Global Outcomes (KDIGO) [8] define four risk categories for the combined prognosis of total mortality, cardiovascular mortality and events related to the flare-up and progression of CKD, but do not specifically mention its importance as a predictive factor of HF.

Treatment with SGLT2 inhibitors has been associated with reduced risk of HF. Currently, all guidelines recommend the use of SGLT2 inhibitors in the presence of HF, especially with reduced ejection fraction [9,10]. Since HF in T2DM patients is a frequent and clinically relevant condition that can be improved with specific treatments, its early detection is essential [11].

Recent international guidelines acknowledge the relevance of HF in DM patients and recommend atrial fibrillation screening [12]. However, beyond the clinical suspicion and classical risk factors monitoring [13], they do not establish specific recommendations on the need to determine natriuretic peptides or perform an echocardiogram.

Our hypothesis was that the probability of a T2DM patient having HF could be predicted from the existence or absence of easily identifiable clinical risk conditions. After natriuretic peptide-based screening, the performance of specific tests (echocardiogram) could confirm the diagnosis and lead to an appropriate treatment.

Our objective, based on the baseline data from the DIABET-IC study (Spanish multicentre study on the prevalence and incidence of heart failure in patients with type 2 diabetes in hospital consultation rooms throughout the country), was to determine the main characteristics associated with the risk of presenting HF, with a special focus on evaluating the usefulness of the risk classification proposed by the KDIGO guidelines.

## 2. Material and Methods

### 2.1. Design

The DIABET-IC study is a multicentre, observational, prospective and analytical study conducted in Spanish hospitals promoted by the Spanish Society of Diabetes (SED) and the Spanish Society of Cardiology (SEC). It has an observational, pragmatic design, with a follow-up of patients under conditions of routine clinical practice. After the initial visit, the planned follow-up was 3 years. The study was conducted in accordance with the Declaration of Helsinki, and it was approved by the Ethics Committee of Toledo Hospital Complex on 28 March 2018 (project identification code 243). All patients signed an informed consent to participate.

This work, with a cross-sectional design, has been conducted with the data obtained from patients in the inclusion visit between 2018 and 2019.

### 2.2. Patients

The inclusion criteria were patients aged 18 years or older, with type 2 diabetes diagnosed at least 1 year before the study start according to the Criteria of the American Diabetes Association (ADA) in force at that time [14]. Patients with type 1 diabetes, with the presence of stage 5 chronic kidney disease or with an estimated life expectancy of less than 3 years due to neoplasms or other serious systemic diseases were excluded.

The subjects were to be seen in external endocrinology and/or cardiology consultation rooms. Each participating centre could include a maximum of 40 patients (20 from endocrinology and 20 from cardiology units). Recruitment was consecutive to avoid selection bias.

The expected sample size was 2400 patients distributed in 60 centres. This sample size, with a confidence level of 95%, would allow the estimation of the incidence of HF for 3 years with an absolute accuracy of 0.6%. However, due to recruitment problems, the desired number of patients was not reached.

### 2.3. Primary Dependent Variable

The main dependent variable analysed was the presence of HF at the inclusion visit. The definition of HF was collected in the guidelines of the European Society of Cardiology (ESC) [15] in 2016. HF was classified as HF with reduced ejection fraction (HFrEF) if the left ventricular ejection fraction (EF) was less than 40%. Due to the limited number of patients, all those with EF greater than or equal to 40% were considered subjects with preserved EF (HFpEF). In a sensitivity analysis, an EF cut-off point of 50% was selected for the definition of HFpEF.

### 2.4. Variables Obtained at Inclusion

Demographic data (age, sex), lifestyle (smoking status, alcohol consumption) and concomitant pathologies: coronary artery disease (CAD), defined as a history of acute myocardial infarction, revascularization or coronary stenosis > 50%; cerebrovascular disease (CD), defined as a history of stroke or carotid stenosis > 50%; peripheral artery disease (PAD), defined as lower limb artery disease; atherosclerotic cardiovascular disease (ACVD), defined as CAD and/or CD and/or PAD, with classification of the number of affected territories into 3 categories (none, one, or more than one); chronic obstructive pulmonary disease (COPD); obstructive sleep apnoea syndrome (OSA); high blood pressure (HBP), defined as blood pressure > 140/90 mmHg or taking hypotensive drugs; atrial fibrillation (AF); fatty liver disease (FLD); and finally the Charlson score.

Data related to T2DM: disease duration; result of the ocular fundus examination, classifying the findings as normal, simple retinopathy or proliferative retinopathy; presence of neuropathy by clinical diagnosis; presence of nephropathy, defined as glomerular filtration rate (GFR) < 60 mL/min/1.73 m^2^ and/or urinary albumin excretion rate (UAER) ≥ 30 mg/g.

Physical examination: weight and height measurements in light clothing and without footwear, with estimation of body mass index (BMI) in the form of weight (Kg)/height (m)^2^; measurement of systolic and diastolic blood pressure, with the patient seated, after at least 5 min of rest, and with a cuff appropriate to the circumference of the arm.

Fasting blood sample collection to measure: glycaemia (mg/dL); HbA1c (%); lipid profile (total cholesterol, triglycerides, HDL-cholesterol), with non-HDL cholesterol (NHDLC) calculation (total cholesterol—HDL) expressed in mg/dL; creatinine (mg/dL), with determination of GFR using the CKD-EPI formula; UAER in a morning urine sample, expressed in mg/g of creatinine; haemoglobin (g/dL); natriuretic peptides (BNP or NT-proBNP) expressed in pg/mL. Measurements were made in each participating hospital following routine procedures.

Electrocardiogram, with rhythm assessment.

Echocardiogram, with measurement of left ventricular ejection fraction (LVEF). The procedure was conducted in each of the participating centres, and patients were classified in 3 categories: preserved (≥50%), intermediate (40–49%) and reduced (<40%), with subsequent recoding into 2 categories: preserved (≥40%) or reduced (<40%).

Patients were also classified based on their GFR (expressed in mL/min/1.73 m^2^) and UAER (expressed in mg/g of creatinine) in 4 risk categories, following the indications of the KDIGO guideline [8]:

Low risk: GFR ≥ 60 and UAER < 30.

Moderate risk: GFR ≥ 60 and UAER 30–300 or GFR 45–59 and UAER < 30.

High risk: GFR ≥ 60 and UAER > 300 or GFR 45-59 and UAER 30–300 or GFR 30–44 and UAER < 30.

Very high risk: GFR 45–59 and UAER > 300 or GFR 30–44 and UAER ≥ 30 or GFR < 30.

### 2.5. Statistical Methods

The quantitative variables were described with their mean and standard deviation (SD) or as median with interquartile range. Qualitative variables were expressed as frequency distribution in %.

A comparison of quantitative variables was made with the student’s t test on independent samples or ANOVA, or with non-parametric tests if the assumptions of normality were violated. The comparison of qualitative variables was done with X^2^ test.

Logistic regression models were performed to determine the risk conferred by the different variables to show HF at the time of study inclusion, calculating the odds ratio (OR) with its 95% confidence interval (CI). Univariate and multivariate analyses were performed. The criteria for the inclusion of the variables in the models were based on their significance in the univariate analysis (*p* < 0.05) or on their clinical meaning. Highly correlated variables were not included simultaneously.

The initial multivariate model included age, sex, COPD, smoking status, HBP (or alternatively systolic and diastolic blood pressure), NHDLC, BMI, OSA, FLD, T2DM duration, presence of retinopathy, HbA1c, haemoglobin, KDIGO risk classification, (or alternatively GFR and UAER), AF, CAD, CD and PAD (or alternatively the number of vascular territories affected). The values of natriuretic peptide tests were not included because they were involved in the diagnostic process. Finally, a sequential exclusion procedure was performed to obtain the independent risk factors of presenting HF.

In a sensitivity analysis, we included in the multivariate models a variable named “origin of the patients.”

Associations with a *p* < 0.05 were considered significant.

## 3. Results

A total of 1517 patients from 58 Spanish hospitals were included, with a mean age of 67.3 (SD 10) years, of which 33% were women. The mean T2DM duration was 14 (SD 11) years, with 42.6% of patients receiving insulin treatment. The proportion of patients with chronic complications was: 12.5% retinopathy (7.6% simple and 4.9% proliferative), 30.3% nephropathy, 5.6% neuropathy and 49% ACVD (38.2% with an affected territory and 10.8% with more than one affected territory). Regarding the KDIGO risk categories, 56.3% of the patients were low risk, 23.3% moderate, 12% high and 8.4% very high.

The proportion of patients with protective cardiovascular treatments was high: 40.5% on SGLT2 inhibitors, 16.5% on GLP1 receptor agonists, 68.6% on ACEI or ARB, 58.5% on beta-blockers, 21.2% on mineralocorticoid receptor antagonists, 11% on sacubitril-valsartan, 84% on statins, 0.9% on PCSK9 inhibitors, 52.7% on antiplatelet agents and 25.7% on oral anticoagulants.

Valid data were obtained for the HF classification in 1497 patients. The prevalence of HF was 37% (16% EF < 40%; 8% EF 40-49%; 13% EF ≥ 50%). The prevalence of HF increased statistically significantly in patients with OSA, in patients with AF, in the presence of ACVD (especially in patients with PAD and as the number of affected territories increased) and in the successive risk categories according to the KDIGO classification (Figure 1). The prevalence was higher than 50% in subjects with AF (66.7%), in the KDIGO very high-risk category (66%), with an atherosclerotic involvement of more than 1 vascular territory (56.7%) and with the presence of PAD (52.4%). The Appendix A shows the prevalence of HF with EF < 40% and ≥40% separately.

As expected, the prevalence of HF was higher in patients coming from the cardiology units than in those coming from the endocrinology units (51.6% vs 16%; *p* < 0.001). The distribution of patients with EF < 40% and ≥40% was different between both units: 25.1% and 26.5%, respectively, in cardiology vs 2.9% and 13.1%, respectively, in endocrinology (*p* < 0.001). There was also a higher proportion of patients with ACVD (61.2% vs 31.7%; *p* < 0.001) in the cardiology sample.

Table 1 compares the patients’ characteristics based on whether they had HF at baseline. Patients with HF were older; had a higher prevalence of COPD, OSA, ACVD and AF; and a lower prevalence of FLD. They also had lower blood pressure and lower NHDLC, haemoglobin and GFR levels, but higher values of UAER and NT-proBNP. Interestingly, patients with HF were characterized by having a greater number of vascular territories affected by atherosclerosis, and a higher proportion of subjects were in the KDIGO high- and very high-risk categories. Table 2 further compares the patients with HF depending on whether their EF is less than 40%. The Appendix A uses the alternative cut-off point of 50%.

In the multivariate analyses, the independent predictors of the presence of HF (Table 3) were increased age (OR per 1 year 1.02; *p* = 0.006), decreased systolic blood pressure (OR per 1 mmHg = 0.98; *p* < 0.001) and haemoglobin (OR per 1 g/dL = 0.86; *p* < 0.001), the presence of OSA (OR = 1.61; *p* = 0.006), KDIGO risk classification (high-risk OR = 2.46 and very high-risk OR = 3.39; *p* < 0.001 both) and the presence of ACVD (OR 1 territory = 1.38 and OR more than one territory = 2.39; *p* = 0.02 and *p* < 0.001 respectively). Conversely, the presence of FLD was a protective factor in the limit of statistical significance (OR = 0.59; *p* = 0.016). The statistical significance of FLD was lost after adjusting for origin of the patients (OR = 0.83; *p* = 0.41).

Figure 2 shows the distribution of patients and the adjusted OR for the prevalence of HF in the four KDIGO risk categories.

## 4. Discussion

In our work, conducted in a patient sample with T2DM coming from outpatient cardiology and endocrinology units, with a prevalence of HF of 37%, we found that the main clinical conditions independently associated with the presence of HF were age, the presence of OSA, atherosclerotic CVD and renal involvement.

The prevalence of HF has increased in recent years, justified by the aging of the population and better survival after diagnosis [16]. T2DM and HF often coexist, and each disease increases the risk of the onset of the other [17]. In a meta-analysis of 31 studies with more than 41,000 patients, the prevalence of T2DM in patients with HF was 23%, similar to the prevalence in patients with preserved or reduced EF [18].

Similarly, in a meta-analysis of cardiovascular safety trials in patients with T2DM, the prevalence of HF has been detected to be between 9% and 28%, although without systematic performance of natriuretic peptide tests or echocardiogram [19]. The prevalence obtained in our study, 37%, should be interpreted considering the origin of the patients and the performance of systematic screening. Although the sample derived from the cardiology units would be clearly biased toward obtaining high values, the prevalence of 16% obtained in patients recruited in the endocrinology units may be valid for subjects in the sixth decade of life and with a duration of the disease greater than 10 years. The higher proportion of HFrEF in cardiology patients can be justified by their greater presence of ACVD.

On the other hand, the number of comorbidities associated with the presence of HF has been described as high [16], ranging from 3.4 in 2002 to 5.4 in 2014. In our study, patients with HF, in addition to T2DM, had a higher prevalence of HBP (although with lower systolic blood pressure, especially in the case of HFrEF, justified by the greater intensity of pharmacological treatment and/or deterioration of ventricular function), COPD, OSA, AF, atherosclerotic vascular involvement in any vascular territory and renal impairment.

Patients with OSA are characterized by higher blood pressure, insulin resistance and activity of the sympathetic nervous system, which favours the appearance of arrhythmias and HF [20]. In addition, they also have a higher prevalence of visceral obesity, which has harmful effects on the myocardium, favouring inflammation, fibrosis and the development of two strongly related pathologies, AF and HFpEF [21]. In our study, the presence of OSA increased the prevalence of HF by approximately 50% and the presence of AF increased risk by almost a factor of 5, although this association was not statistically significant in the multivariate analysis.

PAD is the manifestation of ACVD most strongly associated with DM, and its presence may increase the risk of major cardiovascular events (MACE) even more than CAD [22] itself. In the DECLARE [23] study, patients with PAD had a higher prevalence of HF and a higher incidence of hospitalization for HF. More importantly, in both the FOURIER [24] and EMPA-REG OUTCOME [25] studies, the atherosclerotic involvement of multiple vascular territories has been shown to increase the risk of MACE and HF more than that of a single one. In our work, we found that the involvement of more than 1 vascular territory doubled the prevalence of HF.

The last aspect, and probably the most important, was that we found a great impact of the KDIGO risk classification (high- and very high-risk levels) on the prevalence of HF. This finding is biologically plausible [7], and has been supported by the observation that, in cardiovascular safety studies in type 2 diabetes enriched with patients with CKD, the incidence of HF hospitalization exceeds that of ACVD [19]. In a secondary analysis of the CANVAS [26] study, there was also an increase in the prevalence of HF and in the incidence rate of hospitalization for HF in the successive KDIGO risk categories, with the beneficial effect of canagliflozin being more marked in absolute terms in the highest risk categories. Given the association between HF and CKD, the KDIGO guidelines recommend developing strategies to diagnose and treat both conditions at an early stage [27]. Importantly, the additive effect of polyvascular disease and CKD found in our study on HF risk has also been confirmed in other populations [25]. Our additional finding that haemoglobin was a protective factor can be justified by its close relationship with renal function.

It was surprising to find a lower prevalence of FLD in patients with HF, since the association between hepatic fibrosis and diastolic dysfunction has been described [28]. However, in our sample, no imaging tests or biomarker panels were systematically performed, so the collected prevalence of FLD (12.1%) was much lower than that described [29] in patients with T2DM, up to 70%. Furthermore, the statistical significance of FLD was lost after adjusting for the origin of the patients. 

As the strengths of our work, we highlight the recruitment of a large sample of T2DM patients, the systematic performance of natriuretic peptide tests and echocardiograms to accurately diagnose the prevalence of HF, and the collection of multiple demographics, clinical and analytical variables to evaluate factors associated with the presence of HF. As for the weaknesses, it should be noted that the origin of the patients implies a selection bias, that the desired sample size was not reached due to recruitment problems, that the cross-sectional design did not allow the establishment of the causal effect of the associations found and that the multicentre design did not allow the standardization of biochemical and echocardiographic measurements. In fact, the associations of FLD and lower blood pressures with HF may be related to which clinic the diabetes patients came from and not potential etiologic associations.

In conclusion, we found that the KDIGO risk classification (high- and very high-risk categories), the presence of OSA and the existence of ACVD (especially in the case of polyvascular disease) are useful to predict the presence of HF in T2DM patients. Therefore, we consider it necessary to perform a systematic screening of HF in T2DM patients who meet these criteria to administer the appropriate treatment and improve the prognosis.

## Figures and Tables

**Figure 1 jcm-10-04634-f001:**
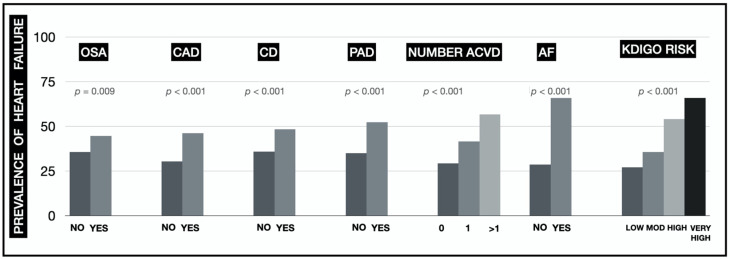
Prevalence of HF based on the patients’ concomitant conditions (OSA: obstructive sleep apnoea syndrome; CAD: coronary artery disease; CD: cerebrovascular disease; PAD: peripheral artery disease; ACVD: atherosclerotic cardiovascular disease; AF: atrial fibrillation; KDIGO: Kidney Disease Improving Global Outcomes).

**Figure 2 jcm-10-04634-f002:**
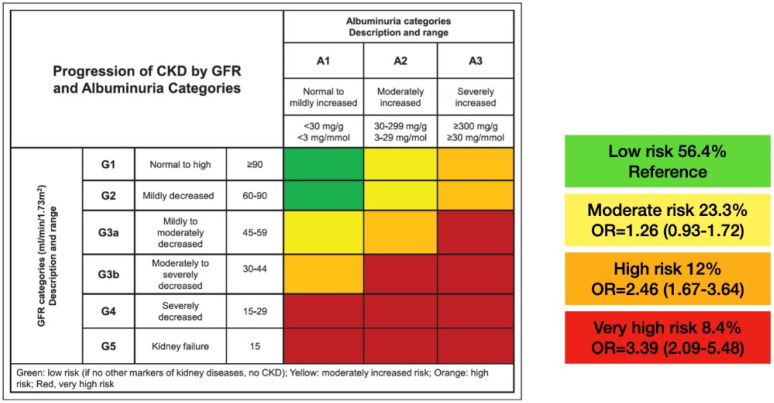
Distribution of the sample into the KDIGO categories and adjusted OR for the risk of having HF at the inclusion visit. CKD: chronic kidney disease; GFR: glomerular filtration rate; G: glomerular filtration rate categories; A: albuminuria categories; OR: odds ratio.

**Table 1 jcm-10-04634-t001:** Patients’ baseline characteristics and univariate OR for the risk of having HF.

Variable		Total Group *n* = 1497	No Heart Failure *n* = 943	Heart Failure *n* = 554	*p*	OR	95% CI
Age (years)		67.3 (10.1)	65.9 (9.9)	69.8 (9,9)	<0.001	1.04	1.03–1.05
Gender (% female)		501 (33)	320 (33.9)	169 (30.5)	0.17	0.86	0.68–1.07
Current smoking (%)		164 (10.8)	116 (12.3)	46 (8.3)	0.12	0.74	0.51–1.08
Hypertension (%)		1244 (81.9)	763 (80.9)	466 (84.1)	0.12	1.25	0.94–1.65
Chronic obstructive pulmonary disease (%)		166 (10.9)	79 (8.4)	85 (15.3)	<0.001	1.98	1.43–2.75
Obstructive sleep apnoea (%)		232 (15.3)	126 (13.4)	102 (18.4)	0.009	1.46	1.10–1.95
Fatty liver disease (%)		181 (12.1)	132 (14.2)	43 (7.8)	<0.001	0.51	0.35–0.73
Auricular fibrillation (%)		327 (21.6)	108 (11.4)	217 (39.2)	<0.001	4.98	3.83–6.48
Coronary artery disease (%)		633 (41.9)	339 (36)	291 (52.5)	<0.001	1.97	1.59–2.44
Cerebrovascular disease (%)		128 (8.5)	66 (7)	62 (11.2)	<0.001	1.67	1.16–2.41
Peripheral artery disease (%)		164 (10.8)	78 (8.3)	86 (15.5)	<0.001	2.04	1.47–2.82
Atherosclerotic cardiovascular disease (%)		741 (49)	406 (43)	332 (59.9)	<0.001	1.98	1.60–2.45
Number of vascular territories affected	0 (%)	771 (51)	537 (57)	222 (40.1)	<0.001	1	-
	1 (%)	577 (38.2)	335 (35.5)	239 (43.1)		1.73	1.37–2.17
	>1 (%)	164 (10.8)	71 (7.5)	93 (16.8)		3.17	2.24–4.48
Charlson index (points)		0.62 (0.88)	0.52 (0.8)	0,81 (0.98)	<0.001	1.44	1.28–1.62
Retinopathy	Simple (%)	114 (7.6)	77 (8.2)	32 (5.8)	0.29	0.69	0.45–1.05
	Proliferative (%)	74 (4.9)	45 (4.8)	29 (5.2)		1.07	0.66–1.72
Body mass index (Kg/m^2^)		30.3 (5.2)	30.3 (5.1)	30.1 (5.4)	0.57	1.00	0.98–1.02
Systolic blood pressure (mmHg)		135 (19.4)	138 (18)	129 (21)	<0.001	0.98	0.97–0.98
Diastolic blood pressure (mmHg)		76 (11.5)	77 (11)	73 (11)	<0.001	0.96	0.95–0.97
Diabetes duration (years)		14 (11.1)	13.7 (9.9)	14.7 (12.9)	0.14	1.01	1.00–1.02
HbA1c (%)		7.3 (1.3)	7.3 (1.3)	7.2 (1.3)	0.12	0.94	0.86–1.02
Non-HDL cholesterol (mg/dL)		109 (34.1)	110 (34)	107 (35)	0.04	0.99	0.99–1.00
Triglycerides (mg/dL) *		133 (85)	129 (87)	136 (82)	0.55	1.00	0.99–1.00
Haemoglobin (g/dL)		14 (1.8)	14.3 (1.7)	13.6 (1.9)	<0.001	0.82	0.77–0.87
NT-proBNP (pg/mL) *		243 (782)	127 (245)	851 (1958)	<0.001	1.001	1.000–1.002
Glomerular filtration rate (mL/min/1.73 m^2^)		73.1 (22.7)	77.9(20.8)	64.6 (23.5)	<0.001	0.97	0.96–0.98
Urinary albumin excretion (mg/g)		87 (393)	62 (201)	132 (603)	0.017	1.001	1.000–1.001
KDIGO risk	Low (%)	715 (56.4)	518 (63.6)	194 (43)	<0.001	1	-
	Moderate (%)	295 (23.3)	190 (23.4)	105 (23.3)		1.48	1.10–1.97
	High (%)	152 (12)	70 (8.6)	82 (18.2)		3.13	2.18–4.48
	Very high (%)	106 (8.4)	36 (4.4)	70 (15.5)		5.19	3.36–8.02

The values in the boxes are the number of patients (%), mean (standard deviation) or median * (interquartile range). OR: odds ratio; CI: confidence interval; HbA1c: glycated haemoglobin; HDL: High-density lipoprotein; NT-proBNP: N-terminal fragment of brain natriuretic peptide; KDIGO: Kidney Disease Improving Global Outcomes.

**Table 2 jcm-10-04634-t002:** Patients’ baseline characteristics based on the type of HF.

Variable		Total Group *n* = 1497	Group 1: No Heart Failure *n* = 943	Group 2: HF with EF < 40% *n* = 240	Group 3: HF with EF ≥ 40% *n* = 314	*p*	*p* Group 2 vs 3
Age (years)		67.3 (10.1)	65.9 (9.9)	68.1 (9.9)	71.1 (9.7)	<0.001	<0.001
Gender (% female)		501 (33)	320 (33.9)	52 (21.7)	117 (37.2)	<0.001	<0.001
Current smoking (%)		164 (10.8)	116 (12.3)	27 (11.2)	19 (6.1)	<0.001	0.017
Hypertension (%)		1244 (81.9)	763 (80.9)	188 (78.3)	278 (88.5)	0.002	0.001
Chronic obstructive pulmonary disease (%)		166 (10.9)	79 (8.4)	39 (16.2)	46 (14.6)	<0.001	0.60
Obstructive sleep apnoea (%)		232 (15.3)	126 (13.4)	37 (15.4)	65 (20.7)	0.007	0.11
Fatty liver disease (%)		181 (12.1)	132 (14.2)	14 (5.9)	29 (9.3)	<0.001	0.14
Auricular fibrillation (%)		327 (21.6)	108 (11.4)	92 (38.3)	125 (39.8)	<0.001	0.72
Coronary artery disease (%)		633 (41.9)	339 (36)	148 (61.6)	143 (45.5)	<0.001	<0.001
Cerebrovascular disease (%)		128 (8.5)	66 (7)	28 (11.7)	34 (10.8)	0.019	0.76
Peripheral artery disease (%)		164 (10.8)	78 (8.3)	37 (15.4)	49 (15.6)	<0.001	0.95
Atherosclerotic cardiovascular disease (%)		741 (49)	406 (43)	164 (68.3)	168 (53.5)	<0.001	<0.001
Number of vascular territories affected	0	771 (51)	537 (57)	76 (31.7)	146 (46.5)	<0.001	0.002
	1	577 (38.2)	335 (35.5)	120 (50)	119 (37.9)		
	>1	164 (10.8)	71 (7.5)	44 (18.3)	49 (15.6)		
Charlson index (points)		0.62 (0.88)	0.52 (0.8)	0.78 (0.95)	0.83 (1.00)	<0.001	0.50
Retinopathy	Simple (%)	114 (7.6)	77 (8.2)	14 (5.8)	18 (5.7)	0.64	0.88
	Proliferative (%)	74 (4.9)	45 (4.8)	13 (5.4)	16 (5.1)		
Body mass index (Kg/m^2^)		30.3 (5.2)	30.3 (5.1)	29.4 (5.3)	30.7 (5.4)	0.012	0.005
Systolic blood pressure (mmHg)		135 (19.4)	138 (18)	126 (19.8)	132 (19.8)	<0.001	<0.001
Diastolic blood pressure (mmHg)		76 (11.5)	77 (11)	72 (11.8)	73 (11.2)	<0.001	0.10
Diabetes duration (years)		14 (11.1)	13.7 (9.9)	15.4 (15.3)	14.1 (10.7)	0.1	0.46
HbA1c (%)		7.3 (1.3)	7.3 (1.3)	7.2 (1.2)	7.3 (1.3)	0.26	0.58
Non-HDL cholesterol (mg/dL)		109 (34.1)	110 (34)	104 (34)	109 (35)	0.026	0.09
Triglycerides (mg/dL) *		133 (85)	129 (87)	134 (82)	138 (84)	0.33	0.16
Haemoglobin (g/dL)		14 (1.8)	14.3 (1.7)	13.9 (1.9)	13.4 (1.8)	<0.001	0.004
NT-proBNP (pg/mL) *		243 (782)	127 (245)	1089(2661)	627 (1563)	<0.001	0.038
Ejection Fraction (%)		54.9 (13.4)	61.5 (7.6)	32.8 (7.9)	54 (10.4)	<0.001	<0.001
Glomerular filtration rate (mL/min/1.73 m^2^)		73.1 (22.7)	77.9 (20.8)	66.3 (23.1)	66.3 (23.8)	<0.001	0.14
Urinary albumin excretion (mg/g)		87 (393)	62 (201)	165 (825)	112 (404)	0.004	0.36
KDIGO risk	Low (%)	715 (56.4)	518 (63.6)	73 (41.7)	121 (43.8)	<0.001	0.97
	Moderate (%)	295 (23.3)	190 (23.4)	42 (24)	63 (22.8)		
	High (%)	152 (12)	70 (8.6)	33 (18.9)	49 (17.7)		
	Very high (%)	106 (8.4)	36 (4.4)	27 (15.4)	43 (15.6)		

The values in the boxes are the number of patients (%), mean (standard deviation) or median * (interquartile range).

**Table 3 jcm-10-04634-t003:** Independent variables (multivariate analysis) associated with the risk of HF at the inclusion visit.

Variable		OR	95% CI	*p*
Age (years)		1.02	1.01–1.03	0.006
Obstructive sleep apnoea		1.61	1.14–2.25	0.006
Fatty liver disease		0.59	0.39–0.91	0.016
Coronary artery disease		1.66	1.31–2.11	<0.001
Peripheral artery disease		1.53	1.06-2.23	0.025
Number of vascular territories affected	0	1		-
	1	1.38	1.05–1.81	0.02
	>1	2.39	1.59–3.60	<0.001
Systolic blood pressure (mmHg)		0.98	0.97–0.99	<0.001
Haemoglobin (g/dL)		0.86	0.80–0.93	<0.001
Glomerular filtration rate (mL/min/1.73 m^2^)		0.976	0.97–0.98	<0.001
KDIGO risk	Low	1		-
	Moderate	1.26	0.93–1.72	0.14
	High	2.46	1.67–3.64	<0.001
	Very high	3.39	2.09–5.48	<0.001

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
