# Peer review of "Association of the KDIGO Risk Classification with the Prevalence of Heart Failure in Patients with Type 2 Diabetes"

_jcm, 2021, doi:10.3390/jcm10204634_

Round 1
Reviewer 1 Report
This is an interesting paper, that reports the cross-sectional findings of an association of kidney disease (KDIGO) with heart failure in a cohort of hospitalized diabetic patients examined in cardiology and endocrine clinics.
Discussion
This studies limitations need to be highlighted more.
The title and abstract are is misleading using the words , "predicting" suggests a prospective study but this is cross-sectional analysis. Line 21 states that DIABETIC-IC is a prospective study, which it may well be in future years, but the authors present a cross-sectional analysis. The investigators present odds ratios of associations 31-39, not predictors. The wording of the results makes interpretation difficult. The authors switch from "presence of" to "decreased ".
Introduction is fine.
Design of the study is good. Cross-sectional is more universally used than transversal.
2.3Primary dependent variable-
Use EF<40% for HFrEF, but would have liked >50% for HFpEF, at least in a sensitivity analysis.
2.4 Variables
Many of the atherosclerotic variables >50% coronary stenosis and carotid stenosis >50% seem generous, usually a 70-80% stenosis is needed.
2.5 stat methods are well described.
Since participants came from endocrine clinics and cardiology clinics, a clinic variable should be added to the models or use IPW to account for differential effects of those with diabetes with HF and those not. The association that hepatic steatosis is related reduced risk of HF and lower blood pressures with HF may be related to which clinic the diabetes patients came from and not potential etiologic associations.
Results as presented are fine. Many of the associations appear counter intuitive, increased systolic blood pressure is associated with reduced risk of heart failure. This may be to more aggressive blood pressure treatment or concomittant treatment of HF or selection bias from cardiology clinics vs endocrine clinics.
Table 3 has % under variable names but present odd ratios, the % should be removed.
Author Response
This is an interesting paper, that reports the cross-sectional findings of an association of kidney disease (KDIGO) with heart failure in a cohort of hospitalized diabetic patients examined in cardiology and endocrine clinics.
Discussion
This study limitations need to be highlighted more.
Limitations are extended
The title and abstract are is misleading using the words , "predicting" suggests a prospective study but this is cross-sectional analysis. Line 21 states that DIABETIC-IC is a prospective study, which it may well be in future years, but the authors present a cross-sectional analysis. The investigators present odds ratios of associations 31-39, not predictors. The wording of the results makes interpretation difficult. The authors switch from "presence of" to "decreased ".
We have changed in title and abstract the term “predict” by the term “association” or “screen”
Introduction is fine.
Design of the study is good. Cross-sectional is more universally used than transversal.
Cross-sectional is used instead of transversal in abstract and methods
2.3Primary dependent variable-
Use EF<40% for HFrEF, but would have liked >50% for HFpEF, at least in a sensitivity analysis.
A supplementary table is added with a cut-off point of 50%
2.4 Variables
Many of the atherosclerotic variables >50% coronary stenosis and carotid stenosis >50% seem generous, usually a 70-80% stenosis is needed.
We selected 50% stenosis because this is the threshold chosen by ESC/EAS guidelines to define documented ASCVD.
2.5 stat methods are well described.
Since participants came from endocrine clinics and cardiology clinics, a clinic variable should be added to the models or use IPW to account for differential effects of those with diabetes with HF and those not. The association that hepatic steatosis is related reduced risk of HF and lower blood pressures with HF may be related to which clinic the diabetes patients came from and not potential etiologic associations.
This is a very interesting comment. We have performed a sensitivity analysis with inclusion in the multivariate models of a variable “origin of the patients”. The only clinically relevant change was the loss of statistical significance for hepatic steatosis. This fact is included in the methods, results and discussion.
Results as presented are fine. Many of the associations appear counter intuitive, increased systolic blood pressure is associated with reduced risk of heart failure. This may be to more aggressive blood pressure treatment or concomittant treatment of HF or selection bias from cardiology clinics vs endocrine clinics.
This comment is included in discussion
Table 3 has % under variable names but present odd ratios, the % should be removed.
The inclusion of “%” was a mistake and they have been removed.
Reviewer 2 Report
Manuscript Number: jcm_1349521-peer-review
Relevance of the KDIGO risk classification in predicting the prevalence of heart failure in patients with type 2 diabetes.
The purpose of this retrospectively analyzed, prospective study, entitled “Relevance of the KDIGO risk classification in predicting the prevalence of heart failure in patients with type 2 diabetes”, by Gimeno-Orna J.A. et al, was to determine characteristics (excluding natriuretic peptides and echocardiographic examination) associated with the presence of heart failure (HF) in type 2 diabetics (T2D).
The authors found in their diabetic study-population a HF prevalence of 37%, which is slightly higher than what we know from previous trials. The analysis validated known independent predictors for the presence of heart failure as increased age, low systolic blood pressure, decreased hemoglobin, obstructive sleep apnoea, and severity of atherosclerosis. Surprisingly, absence of hepatic steatosis was independently associated with presence of heart failure in this study and a KDIGO high risk, or very high risk classification, also predicted presence of heart failure, respectively.
It is postulated, that in diabetic patients with high and very high risk KDIGO categories, systematic HF screening should be carried out.
All in all, the study conduct seems appropriate and the results appear to be valid.
However, there are critiques and a few specific comments:
Generally:
- Assessment of the prevalence of HF in T2D is interesting, however, when it comes to risk assessment and timely treatment initiation, it would be much more interesting to assess incidence of HF over time. As follow-up visits have been scheduled, the authors should provide information on outcome (as occurrence of HF during follow-up, CV events). Were the identified parameters at baseline of predictive value for future events? This is particularly relevant when considering data on NT-proBNP, which show that the single biomarker is highly performing for the assessment of CV risk in T2D. A easily to obtain single NT-proBNP measure has been shown to be superior to other established multiparametric assessment tools such as SCORE and the ESC/EASD risk stratification model in predicting 10-year CV and fatal events overall. (Prausmüller S. et al; Cardiovasc Diabetol. 2021 Feb 2;20(1):34. doi: 10.1186/s12933-021-01221-w).
- Some paragraphs of the introduction are hard to follow and sound context is missing. What do the authors mean by “the presence of a specific cardiomyopathy”? Please clarify. Moreover, for example insulin resistance, and hyperglycemia are also present in HFrEF. Among others, please revise the entire paragraph.
- Next, when it comes to SGLT2i treatment, the contextual transition is missing, and hence, the part is confusing to read. Please revise.
- Please write out all abbreviations the first time they are mentioned. Moreover, angiotensin receptor blocker (ARB) is more common than ARA2 and should be replaced.
- There are some typos throughout the text, e.g. page 2, line 50 “....the increased risk given by T2DM is not completely neutralized....; page 2, line 50 “with a special focus on...”.
Table.
- The authors need to improve data presentation on the tables. It is not enough to just mention percentages. Absolute values should also be included. And it is necessary to indicate which value is given in the brackets, e.g. age (years) = 67.3(10.1) does not fit.
Figure.
- Summarized as “heart failure”, HFpEF and HFrEF have a lot in common, especially when it comes to symptoms that are complained. However, especially in terms of risk factors and diagnostic evaluation, both need to be treated as almost two different disease entities. Therefore, I recommend including both HFpEF and HFrEF separately in the figure.
- Legends are missing
Author Response
The purpose of this retrospectively analyzed, prospective study, entitled “Relevance of the KDIGO risk classification in predicting the prevalence of heart failure in patients with type 2 diabetes”, by Gimeno-Orna J.A. et al, was to determine characteristics (excluding natriuretic peptides and echocardiographic examination) associated with the presence of heart failure (HF) in type 2 diabetics (T2D).
The authors found in their diabetic study-population a HF prevalence of 37%, which is slightly higher than what we know from previous trials. The analysis validated known independent predictors for the presence of heart failure as increased age, low systolic blood pressure, decreased hemoglobin, obstructive sleep apnoea, and severity of atherosclerosis. Surprisingly, absence of hepatic steatosis was independently associated with presence of heart failure in this study and a KDIGO high risk, or very high risk classification, also predicted presence of heart failure, respectively.
It is postulated, that in diabetic patients with high and very high risk KDIGO categories, systematic HF screening should be carried out.
All in all, the study conduct seems appropriate and the results appear to be valid.
However, there are critiques and a few specific comments:
Generally:
- Assessment of the prevalence of HF in T2D is interesting, however, when it comes to risk assessment and timely treatment initiation, it would be much more interesting to assess incidence of HF over time. As follow-up visits have been scheduled, the authors should provide information on outcome (as occurrence of HF during follow-up, CV events). Were the identified parameters at baseline of predictive value for future events? This is particularly relevant when considering data on NT-proBNP, which show that the single biomarker is highly performing for the assessment of CV risk in T2D. A easily to obtain single NT-proBNP measure has been shown to be superior to other established multiparametric assessment tools such as SCORE and the ESC/EASD risk stratification model in predicting 10-year CV and fatal events overall. (Prausmüller S. et al; Cardiovasc Diabetol. 2021 Feb 2;20(1):34. doi: 10.1186/s12933-021-01221-w).
The DIABET-IC study is a multicentre, observational, prospective, and analytical study, conducted in Spanish hospitals promoted by the Spanish Society of Diabetes (SED) and the Spanish Society of Cardiology (SEC). However, the study is ongoing, and we currently only have baseline data
- Some paragraphs of the introduction are hard to follow and sound context is missing. What do the authors mean by “the presence of a specific cardiomyopathy”? Please clarify. Moreover, for example insulin resistance, and hyperglycemia are also present in HFrEF. Among others, please revise the entire paragraph.
The paragraph has been modified to define the reference to “diabetic cardiomyopathy”
- Next, when it comes to SGLT2i treatment, the contextual transition is missing, and hence, the part is confusing to read. Please revise.
To connect paragraphs, the sentence “Treatment with SGLT2 inhibitors has been associated with reduced risk of HF” has been added.
- Please write out all abbreviations the first time they are mentioned. Moreover, angiotensin receptor blocker (ARB) is more common than ARA2 and should be replaced.
ARB is used instead of ARB
- There are some typos throughout the text, e.g. page 2, line 50 “....the increased risk given by T2DM is not completely neutralized....; page 2, line 50 “with a special focus on...”.
Typos have been corrected
Table.
- The authors need to improve data presentation on the tables. It is not enough to just mention percentages. Absolute values should also be included. And it is necessary to indicate which value is given in the brackets, e.g. age (years) = 67.3(10.1) does not fit.
Number of patients are included in the first row. In legend of the table is defined the value of the boxes.
Figure.
- Summarized as “heart failure”, HFpEF and HFrEF have a lot in common, especially when it comes to symptoms that are complained. However, especially in terms of risk factors and diagnostic evaluation, both need to be treated as almost two different disease entities. Therefore, I recommend including both HFpEF and HFrEF separately in the figure.
A supplementary figure is added with HFpEF and HFrEF separately.
- Legends are missing
Legends of figures are included in the text
Round 2
Reviewer 1 Report
The revised paper has responded to all my concerns
Reviewer 2 Report
Thanks to the authors for submitting their revised version of the manuscript entitled "Association of the KDIGO risk classification with the prevalence of heart failure in patients with type 2 diabetes" with Gimeno-Orna JA as first author.
Referring to my previous comments:
- Table: Except for the legends, the table has not been adjusted (absolute and relative values) as stated by the author? Was the old version provided accidentally?
- Supplemental figure: A supplementary figure illustrating HFpEF and HFrEF separately has been added. However, to the best of my understanding, association with comorbidities/risk factors has not been calculated separately for both disease entities. I assume the given p-value refers to the overall analysis? However, in order to be able to relate the risk factors to the respective heart failure entity, the authors should provide analysis for both HFpEF and HFrEF separately. Furthermore, the figure would improve when the bars were grouped according to the respective HF unit.